# Recent Computational Advances Regarding Amyloid-β and Tau Membrane Interactions in Alzheimer’s Disease

**DOI:** 10.3390/molecules28207080

**Published:** 2023-10-13

**Authors:** Phuong H. Nguyen, Philippe Derreumaux

**Affiliations:** 1CNRS, Laboratoire de Biochimie Théorique, Institut de Biologie Physico-Chimique, Fondation Edmond de Rothschild, Université Paris Cité, UPR 9080, 13 rue Pierre et Marie Curie, 75005 Paris, France; phuong.nguyen@ibpc.fr; 2Institut Universitaire de France (IUF), 75005 Paris, France

**Keywords:** amyloid, aggregation, simulations, Alzheimer’s disease, membrane

## Abstract

The interactions of amyloid proteins with membranes have been subject to many experimental and computational studies, as these interactions contribute in part to neurodegenerative diseases. In this review, we report on recent simulations that have focused on the adsorption and insertion modes of amyloid-β and tau proteins in membranes. The atomistic-resolution characterization of the conformational changes of these amyloid proteins upon lipid cell membrane and free lipid interactions is of interest to rationally design drugs targeting transient oligomers in Alzheimer’s disease.

## 1. Introduction

Amyloid-β (Aβ) proteins of residues 40 and 42 and tau isoforms of residues 352–421, which are linked to Alzheimer’s disease (AD), have the propensity to aggregate into amyloid fibrils with a cross-β structure [1,2] and multiple polymorphs, as observed by Scheres et al. and Tycko et al. [3,4]. Aggregation proceeds through primary and secondary (surface catalysis and fragmentation) nucleation mechanisms, fibril growth, and depolymerization in various experimental conditions [5].

The Aβ42 sequence consists of hydrophilic, charged, N-terminal residues 1–16; the central hydrophobic core (CHC, residues 17–21); a hydrophilic region (residues 22–28) and hydrophobic residues 30–42. Numerous studies have reported that low-molecular-weight (LMW) oligomers of Aβ generated during the lag phase of aggregation or released from the fibrils are the key players in AD [1,6,7]. The aggregation kinetics of Aβ42 in aqueous solution has been continuously studied to understand the impact of pH variation [8], N-terminal residues [9], and mixing with C-terminal truncated species such as Aβ24 [10]. The computational structure characterization of Aβ42 and tau monomer/dimer in aqueous solution has been pursued [11,12,13,14,15].

Tau oligomers are toxic in vitro, seeding the propagation of AD pathology [16], and the tau 297–391 fragments form fibrils under physiological conditions similar to those found in AD brain tissues [17]. Single immunogold-labeled transmission electron microscope (TEM) and fluorescence spectroscopy revealed the presence of a tyrosine–tyrosine crosslink at position 310 on AD-brain-derived tau oligomers and fibrils [18].

The etiology of AD is linked in part to the mediated effects of Aβ and tau on neuronal cell membranes. Atomic force spectroscopy (AFM) revealed that the self-assembly of Aβ was enhanced and occurred at nanomolecular concentrations on two lipid bilayers formed by 1-palmitoyl-2-oleoyl-glycero-3-phosphocholine (POPC, PC) and 1-palmitoyl-2-oleoyl-sn-glycero-3-phospho-l-serine (POPS, PS) [19]. The presence of cholesterol in the membrane enhanced the primary nucleation of Aβ42 by several orders of magnitude compared with that in aqueous solution, as assessed using thioflavin T (ThT) fluorescence [20]. The experimental techniques used to study the amyloid–membrane interactions through the carpeting, detergent-like removal of lipids, and amyloid–pore hypotheses and to study experimental imaging Aβ membrane interactions were recently reviewed [21,22].

Structures of Aβ–membrane assemblies are challenging experimentally as they are metastable and heterogeneous. Among the new experimental results, using mass spectrometry and nuclear magnetic resonance (NMR) spectroscopy in docecylphosphocholine (DPC) micelles, Carulla et al., at pH 9.0, provided the high-resolution structure of the Aβ42 tetramer and octamer, revealing conductivity pores as a mechanism for the damage of the membrane. The tetramer conformation comprises a six-stranded β-sheet core with the two external Aβ peptides forming β hairpins (Figure 1) [23]. Cryoelectron tomography was used to obtain 3D nanoscale images of the interactions of various Aβ assemblies with liposomes. Aβ oligomers and protofibrils bind to the vesicles, and insert and carpet the upper leaflet of the bilayer, allowing oligomers to establish a network of Aβ linked to liposomes. The monomeric and fibrillar Aβ species have almost no impact on the membrane [24]. It was also evidenced that the Aβ oligomers generated during secondary surface nucleation disrupt membrane integrity, and time-resolved fluorescence microscopy of individual synthetic vesicles demonstrated that the disruption of lipid bilayers correlates with the levels of secondary nucleation-generated Aβ42 oligomers [25]. Experiments also showed that ganglioside-enriched vesicles promoted Aβ oligomerization and disruption of the membrane [26]. The impact of metal ions or lipid vesicles on Aβ aggregation has been studied [2,27,28]; their combined effects are not well understood. It was shown, however, that the co-effect of Cu^2+^ and lipid vesicles led to the rapid formation of abundant Aβ40 LMW oligomers containing β-sheet-rich structures [29].

Surface-enhanced infrared (IR) absorption spectroscopy and atomic force microscopy (AFM) were used to study the structural morphology of Aβ42 aggregation on distinct two-dimensional self-assembled monolayers (SAMs). On hydrophobic interfaces, the interactions with the C terminus of Aβ42 promote the formation of small oligomers with low contents of β-sheet structures. On hydrophilic interfaces, H-bond and electrostatic interactions result in the promotion of large oligomers with a β-sheet structure [30]. Solid-state NMR (ss-NMR) was used to investigate the interactions between Aβ40 peptides and synaptic plasma membranes extracted from rat brain tissues. It was found that, at the membrane-associated nucleation stage, the G25-V36 residues initialized the assembling of low-order β-sheet structures, and the F19-G25 residues were closely linked with lipids [31]. Using high speed AFM and nanoscale infrared spectroscopy (IR) on Aβ42 L34T and G37C variants with a POPC/sphingomyelin (SM)/cholesterol/monosialotetrahexosyl-ganglioside (GM1) membrane, Molinari et al. showed that membrane disruption was associated to small oligomers with an antiparallel β-sheet structure [32]. It is worth noting that nonequilibrium flowing conditions promote the aggregation of Aβ on 1,2-dimyristoyl-sn-glycero-3-phosphocholine (DMPC) bilayers [33].

Various studies have reported the presence of both extracellular and intracellular Aβ assemblies. Using large unilamellar vesicles to model different locations of membranes, including the inner leaflet (IL) and outer leaflet (OL) of the plasma membrane, late endosomes, the endoplasmatic reticulum (ER), and the Golgi apparatus, Aβ42 aggregation was monitored using fibril mass concentration as a function of time. Aβ42 aggregation enhancement was observed in IL, OL, and late endosomes. The membrane in the inner leaflet consisted of 22% PC, 40% PE, 26% PS, and 12% cholesterol (chol); the membrane in the outer leaflet consisted of 30% PC, 5% PS, 35% SMm and 30% chol; and the membrane in late endosomes consisted of 40% PC, 15% PE, 7% SM, 23% chol, and 11% bis(monoacyl-glycero)phosphate (BMP). Aβ42 aggregation inhibition was observed in the Golgi membrane, consisting of 43% PC, 17% phosphatidylethanolamine (PE), 9% phosphatidylinositol (PI), 17% SM, and 14% chol; and in the ER membrane, consisting of 52% PC, 26% PE, 9% PI, and 13% chol [34].

In comparison with Aβ, the number of experimental studies on LMW tau oligomers-membranes is fairly small. It is known that tau oligomers interact with the plasma membrane via multiple binding modes with the N-terminal projection domain (residues 1–243 containing a proline-rich region), and the R1−R4 repeats (residues 244–369) [1,35]. Tau interacts with phospholipid tails, facilitating the formation of stable protein–lipid complexes, facilitating cell-to-cell transport [36]. Like Aβ oligomers, tau oligomers can remodel and disrupt membranes, extract lipids, and form pores, and these effects are modulated by protein concentrations and lipid compositions [37,38,39].

The aim of this review is to report on recent computational studies starting the year 2020, aimed at understanding the effects of Aβ40 and Aβ42 alloforms and long tau isoforms on and in cell membranes and vice versa, and the interactions between free lipids and Aβ peptides.

## 2. Adsorption and Insertion of Amyloid-β

Determining the structures of Aβ–membrane assemblies is a computational challenge. The interaction of an Aβ monomer with membranes was investigated via a series of simulations. Using a CHARMM36m force field and molecular dynamics (MD) simulation, Strodel et al. showed the partial insertion of an Aβ42/POPC complex into a 1,2-dioleoyl-sn-glycero-3-phosphocholine (DOPC) bilayer and a deeper insertion of parts of Aβ42 compared with a single peptide [40]. Alonso et al. investigated the interactions of Aβ42 monomers with lipid bilayers consisting of POPC and small amounts (1–5 mol%) of GM1. Using MD simulations with the GROMOS force field, isothermal calorimetry, and Langmuir balance experiments, they showed that Aβ42 adsorbs, and cannot insert into, GM1-containing phospholipid membranes in the liquid-disordered (Ld) state. They also found that GM1, up to 3 mol%, increased Aβ42 binding to the bilayer [41]. Using the same experimental techniques and a membrane consisting of 47.5% POPC, 47.5% SM, and 5% GM1, they found that Aβ42 monomer, but also oligomers and to a lesser extent fibril preparations, inserted into membranes in the liquid-ordered (Lo) state [42].

Hamiltonian replica exchange molecular dynamics simulations with the OPLS force field were performed to determine the impact of oxidized (ox) G25, G29, and G33 residues on monomeric Aβ42 interacting with a membrane made of 70% POPC, 25% cholesterol, and 5% GM1. Owen et al. found that Aβ42 wild-type (WT) and Aβ42-oxG25 peptides have a lower tendency to bind to GM1 than Aβ42-oxG29 and Aβ42-oxG33, and the WT monomer has a higher propensity to insert into the membrane than the three oxidized alloforms [43]. A 2.5 μs MD simulation of Aβ42-Cu^2+^ with a DMPC bilayer revealed that the presence of calcium ions mediated membrane perturbation and inhibited the penetration of the Aβ42 monomer into the membrane [44]. Interestingly, the decrease in the bending rigidity of the POPC membrane, as a result of the incorporation of Aβ40 and Aβ42 monomers, was observed using MD studies and experimental flicker-noise techniques [45]. Finally, Localito et al. studied the interaction of monomeric alpha-helical Aβ40 (WT, H13P, H14P, L17P, and F20P) using three replicates with a single DEPC lipid. From their simulations, they hypothesized that the alpha-helix is a fundamental requirement to fulfill the lipid-chaperon model [46].

The dynamics of Aβ dimers at the surface or inserted into multiple membrane models were also explored. Chang et al. used atomistic Brownian dynamics simulations to study the association of two Aβ42 monomers on SAM surfaces. The conformations included the fibril-like β-rich structure and a random coil structure with a low alpha-helix content. Four SAM surfaces were used, including undecanethiol (hydrophobic, referred to as CH_3_-SAM), 11-mercapto-1-undecanol (hydrophilic, OH-SAM), 11-amino-1-undecanethiol (cationic, NH^+^-SAM), and 11-mercaptoundecanoic acid (anionic, COO^−^-SAM) on a gold substrate. Monomers started to diffuse from a random position at a height of 40 nm above the SAM surfaces. A total of 1000 trajectories for each rigid-body protein conformation was performed to calculate the association and residence times on the four SAM surfaces. For three SAM surfaces, the exception being NH^+^-SAM, the averaged monomer–dimer association time was in the order of 9–10 μs for the β-rich conformation and 5 μs for the random coil conformation, indicating a 5–35% decrease in association time due to the presence of a surface. The residence time was found to vary from 7.4 (OH-SAM) to 8.8 (CH3-SAM) μs. The monomer–dimer association time on NH^+^-SAM with heterogeneous charge distribution could not be assessed as the monomers were not able to associate, because they were strongly bound to the SAM. The addition of Na^+^ greatly enhanced the diffusion on the COO^-^-SAM. This finding emphasizes the importance of considering the role of Na^+^ in vitro [47].

In a second μs MD simulation, Lyubchenko et al. examined the formation of the Aβ42 dimer modeled using the ff99sb-ildn force field with a POPC bilayer. They revealed a β-sheet content of 10% and an alpha-helix content fluctuating between 10 and 35%, with the N-terminal residues and residues 29–31 having the highest propensity to interact. Of interest is that the on-surface aggregation process is dynamic, and oligomers assembled on the surface can dissociate into the solution. These results suggest that on-surface aggregation is one mechanism through which Aβ42 oligomeric species in solution are produced [19].

In a third μs MD simulation, Strodel et al. compared Aβ42 dimerization, modeled using the CHARMM36m force field, in aqueous solution and at a lipid bilayer surface mimicking a neuronal membrane composed of 38% PC, 24% PE, 5% PS, 20% chol, 9% SM, and 4% GM1. They found that the neuronal membrane reduces the dynamics of membrane-bound Aβ42 and inhibits β-sheet formation (β-sheet content of 28%) due to the hydrogen bonds with the sugar groups of GM1. This result is in contrast to the dimerization in the aqueous phase characterized by a random coil to β-sheet transition, which leads to a β-sheet content of 36%, similar to the content observed in Aβ fibrils. The insertion of the peptides into the hydrophobic region of the membrane was not observed, and the membrane was marginally affected [48].

In a fourth study, Fazli et al. performed three MD simulations of 200 ns each at three temperatures with the OPLS force field for proteins to explore the early interaction steps of two separated parallel Aβ40 peptides with the helix-structure content spanning residues 15–37 partially inserted into a DPPC bilayer. Upon association, the helical region becomes disrupted to two smaller fragments with a kink, and the dimeric Aβ species has two types of distribution in DPPC at 300, 310 and 330 K. In the first distribution, the two peptides remain embedded in top leaflet, while in the second distribution, at least one of the peptides penetrates into the bottom leaflet, leading to the local disordering of the DPPC bilayer [49].

Guo and Wang used atomistic MD simulations for a total of 48 μs to investigate the effects of cholesterol on Aβ42 dimerization in lipid DOPC bilayers with different molar ratios of cholesterol (0, 20, and 40 mol %). Cholesterol reduces the time for the formation of stable dimers and has two effects on Aβ−membrane interactions. Firstly, cholesterol enhances the extraction of the C-terminal region from the membrane to the bulk solution. At the ratios of 0 and 20 mol %, Aβ dimers are attached at the membrane−water interface, and at a ratio of 40 mol %, they are repelled to water. Secondly, cholesterol reduces Aβ−membrane interactions, thereby augmenting dimeric interactions [50].

Other coarse-grained (CG) 120 μs MD simulations, performed by Parisini et al., demonstrated the interactions between trimeric or hexameric Aβ40 fibrils with a 100% DPPC bilayer, a 70% DPPC-30% chol bilayer, or a 50% DPPC-50% chol bilayer. The spontaneous binding of the Aβ40 fibrils to the membranes was captured with the CHC (residues 17–21), the K16 residue, and the C-terminal hydrophobic residues involved in the process. Moreover, they showed that while the Aβ40 fibril does not bind to the 100% DPPC bilayer, its binding affinity to the membrane increases with the amount of cholesterol [51].

Coarse-grained Martini MD simulations, performed by Cheng et al., revealed the binding of Aβ17-42 fibrillar dimers–pentamers to interfacial liquid-ordered (Lo) and liquid-disordered (Ld) regions in phase separated lipid rafts with distinct membrane-bound conformations [52]. CG Martini simulations of several μs, generated by Cruz et al., showed the role of cholesterol in the binding of Aβ11-42 and Aβ17-42 fibrils to lipid bilayers containing 30 and 50% cholesterol by promoting electrostatic interactions [53]. By using atomistic MD simulations, Aβ 12 mers were found to more disrupt a neuronal membrane than a fibril [54], with the peptide and membrane parameterized by the CHARMM36m and CHARMM36 force fields, respectively.

## 3. Detergent Effect of Amyloid-β

Amyloid-β oligomers have a strong detergent effect on lipid bilayers [2]. Cholesterol, a component of membranes, is a risk factor in AD, being present in AD senile plaques with a molar ratio of 1:1 [55], and its level in the brain correlating with the severity of dementia in AD individuals [56].

Nguyen and Derreumaux performed atomistic replica exchange molecular dynamics and replica exchange with solute tempering on the dimer/trimer of Aβ42 with cholesterol at a ratio 1:1 [57,58]. They found that the binding spots of cholesterol essentially involve the CHC and L30-M35 residues. The contribution of D22-K28 and the N-terminus residues was also detected. Cholesterol is rarely inserted into the aggregates. Rather, cholesterol molecules are attached as dimers and trimers at the surface of Aβ42 trimers, proposing that they act as a glue to promote the formation of larger aggregates. The formation of larger Aβ42 aggregates due to the presence of free cholesterol was confirmed using AFM images of Aβ42/cholesterol on a lipid bilayer, and it was shown that these aggregates can accumulate in the bulk solution [59].

The impact of free GM1, abundant in mammalian brains, on the two Aβ isoforms was investigated in a combined NMR-MD study performed by Brooks et al. This study revealed the formation of well-ordered, structurally compact GM1+Aβ complexes, altering Aβ aggregation [60]. The presence of free fatty acids such as lauric acid was found computationally by Hansmann et al. to stabilize ring-like and barrel-shaped Aβ42 oligomers [61]. Similar to the simulation results of Aβ42/cholesterol, simulations of the Aβ42 monomer with 1,2-dimyristoyl*-*sn-glycero*-*3-phosphocholine (DMPC) provided evidence that the unfolded Aβ42 peptide adheres to the lipid cluster rather than embeds into it [62]. MD simulations on the μs time scale revealed that three free POPC lipids trigger a disorder-order (alpha-helix, β-strand) transition in the Aβ42 monomer [48].

## 4. Amyloid-β Pore Formation

Membrane-embedded Aβ species with membrane conductance, which do not display a ThT fluorescence signal, can have a wide range of conformations and oligomer sizes [2]. Some form well-defined pores [63,64,65] made, for instance, of six-stranded β-sheet and β-sandwich structures (Figure 1), and other species are spherical without any evidence of discrete channel or pore formation. Atomistic MD simulations showed that many concentric β barrels are consistent with image-averaged electron micrographs [66,67], revealing the strong role of Aβ β-sheet edge conformations in the permeabilization of the membrane [68]. Interestingly, using MD simulations, quantum mechanics, and experimental techniques, La Rosa et al. showed that Aβ40 can also form alpha-helix channels through the GxxG motif [69].

In a recent study, the dynamics of an S-shaped Aβ42 cross-β hexamer model inserted into a lipid bilayer membrane were explored by Nguyen and Derreumaux using two atomistic MD simulations, each two microseconds long [70]. The initial model was characterized by the CHC and residues 30–42 embedded into a DOPC bilayer membrane (Figure 2). Structural secondary, tertiary, and quaternary rearrangements were observed, leading to two distinct metastable species, hexamer and two trimers, accompanied by membrane disruption and water permeation. The study provides evidence of hexamers and trimers with the N-terminal residues located in the top and bottom leaflets of the membrane and a minimal lifetime of several microseconds. Some conformations, but not the majority, have the CHC and C-terminal hydrophobic residues exposed to the solvent. The MD simulations also showed that residues 1–10 in both trimers of run 1, in trimer 1 of run 2, and residues 1-16 in trimer 2 of run 2 were exposed to the solvent (Figure 3). This result is consistent with many simulations of Aβ42 oligomers partially embedded in the membrane [2]. We did not find evidence of β hairpins in the N terminus, as reported for Aβ42 dimers in DOPC bilayers [71]. Overall, the high solvent accessibility of residues 1-16 may help with understanding the linking of liposomes through Aβ oligomers, as assessed via cryoelectron tomography [22].

## 5. Tau–Membrane Interactions

Zhou et al. reported the computational results of the tau K19 (R1, R3, and R4) monomer bound to a POPC/POPS membrane using the ff14SB force field for proteins [72]. Based on previous electron paramagnetic resonance (EPR) results, which established the presence of one helix in each repeat [73], their biased MD simulations on the microsecond timescale showed that the three amphipathic helices stably bind to a POPC–POPS membrane and revealed an important difference in membrane binding stability between the repeats R1 and R3 and R4. In each amphipathic helix, however, a lysine conserved among the microtubule-binding regions, along with other charged and polar sidechains, interacts with lipid headgroups. Furthermore, a conserved valine along with two other nonpolar sidechains insert into the hydrophobic region of the membrane. These two conserved residues form similar interactions with microtubules as observed from the cryo-EM microtubule-bound structure of tau 202–395 [74]. Zhou et al. proposed that this partial mimicry facilitates the transfer of tau from membranes to microtubules [72].

Nguyen and Derreumaux performed MD simulations of the tau R3-R4 domain monomer at the surface of a DOPC–DOPS lipid bilayer, with residues 306–378 forming the fibril core of full-length tau alloforms in the brain of individuals with AD [75]. The two simulations showed that the surface of the membrane does not induce β-sheet formation and leads to an ensemble of structures very different from those in the bulk solution. They also revealed the dynamic interactions of the membrane-bound state of the tau R3-R4 monomer, allowing insertion of the PHF6 motif (residues 306–311), and residues 312–318 and 376–378.

As tau dimers are observed in cells and in the cerebrospinal fluid of AD patients [76], Nguyen and Derreumaux performed MD simulations on the tau R3-R4 domain dimer starting from a cryo-EM state and a compact globular state (Figure 4). They found distinct insertion depths depending on the initial conformation and transient adsorption of the PHF6 motif on the membrane as the results of the high propensity of PHF6 to form parallel β-sheets, and insertion of the C-terminal R4 region into the membrane (Figure 5) [77]. There are therefore differences between the monomer and dimer results. To what extent the dimer results can be extrapolated to full-length tau remains to be determined, as the R2 repeat interacts with negatively charged vesicles [38].

Using 15 μs coarse-grained Martini MD simulations followed by all-atom (AMBER99sb and Slipid force fields) 200 ns MD simulations with the residues spanning the R2-R3 domain initially adopting the fibril cryo-EM conformation, the transient interactions of tau K18 (containing the four repeats) oligomers (dimer and tetramer) on three lipid rafts (control raft, modified CO-raft0containing GM1 clusters on one leaflet (GM-raft), and modified CO-raft-containing PS clusters on one leaflet (PS-raft)) were investigated. Their raft systems included DPPC, DLPC, and cholesterol in all cases. It was found that tau K18 prefers to bind to the boundary domains (Lod) created by the coexisting Lo and Ld domains in the lipid rafts. The stronger binding of tau K18 oligomers to the GM1 and phosphatidylserine (PS) domains was reported, and K18-induced lipid chain order disruption and tau K18 β-sheet formation were detected. The results suggest that GM1 and PS domains, located in the outer and inner leaflets of the neuronal membranes, respectively, are specific membrane domain targets of tau oligomers, and the Lod domains are not. Clearly, much longer all-atom MD simulations are required to identify more stable β-sheet structures on the raft surfaces [78].

Finally, the interactions of tau fibrils with membrane lipids were studied via a multiscale simulation. Based on the CG Martini protocol followed by all-atom CHARMM36 force field 400 ns MD simulations, Bargava et al. studied the interactions of R3-R4 tau filament consisting of two protofibrils with 14 membrane systems consisting of POPC, 1-palmitoyl-2-oleoyl-sn-glycero-3-phosphatidylethanolamine (POPE), 1-palmitoyl-2-oleoyl-sn-glycero-3-phosphatidyl-glycerol (POPG) along with cholesterol for seven different compositions. Tau proteins do not interact similarly with the zwitterionic lipid membranes compared with the charged lipid membranes, with the negatively charged POPG membranes enhancing the binding propensity of tau fibrils. Cholesterol addition alters tau binding affinity with the membrane. The binding of tau fibril induces the loss of the β sheet of the tau residues, destabilizes β-sheet regions depending on the lipid composition and the percentage of cholesterol concentration, and changes the membrane properties including thickness and order parameters of the tails [79].

## 6. Conclusions

We focused on recent simulations, aimed at understanding the adsorption and insertion modes of the two main amyloid proteins associated with AD in various membrane models. Our experimental and computational knowledge on the aggregation and structures of tau isoforms is increasing at a much slower pace than that of Aβ.

Major challenges were identified in the field. For instance, new experimental methods of biomolecular processes in membranes, spanning the microsecond time resolution and detecting populations of intermediates below the percent range, would be very useful for validating the MD-generated models. The impact of fluid flow on lipid bilayers–amyloid protein interactions should also be investigated both experimentally and computationally, as it is known that cells have extracellular matrices that may have very different behaviors compared to vesicles and planar lipid bilayers in the absence of membrane protein receptors.

Most simulations have started from the alpha-helical and fibrillar states, so repeating the simulations with antiparallel orientations of the peptides and longer timescales would provide more information on the conformational ensemble and its impacts on the membrane. The results of recent computations showed that free lipids (cholesterol, GM1) and fatty acids alter the energy landscape of Aβ42 species, and cholesterol induces the formation of much larger Aβ42 aggregates, which might generate species more difficult to clean from the brain’s interstitial system. More information on the large gap between simulations and in vivo conditions can be found in Ref. [15].

## Figures and Tables

**Figure 1 molecules-28-07080-f001:**
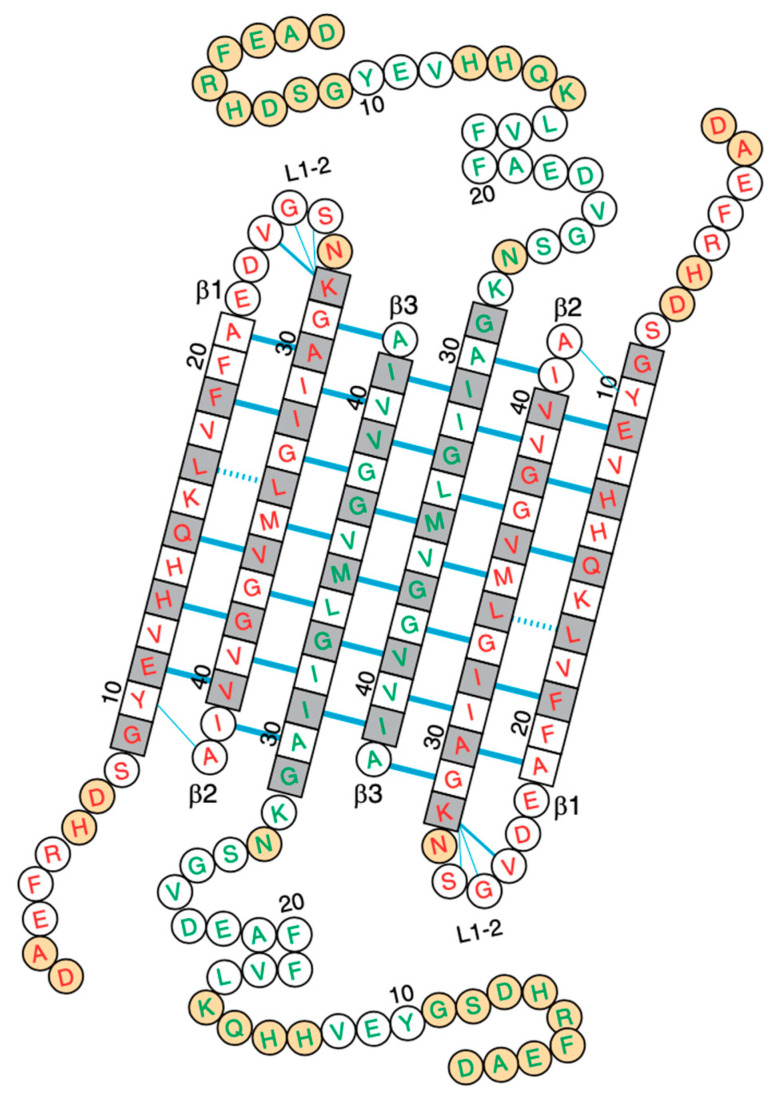
Experimentally derived conformation of the Aβ42 tetramer in a membrane model with a six-stranded β-sheet core and the two external Aβ peptides forming β hairpins, as proposed in Ref. [23].

**Figure 2 molecules-28-07080-f002:**
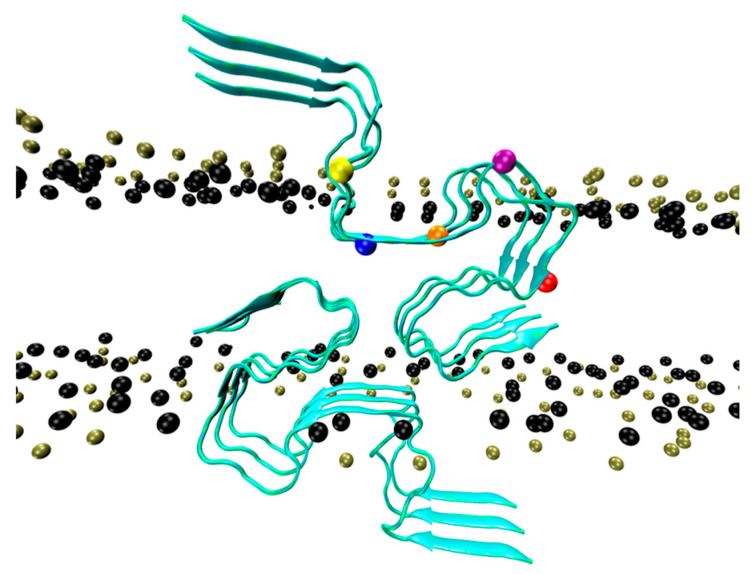
Initial conformation of the Aβ42 hexamer. View in the x direction, parallel to the fibril axis. We show the ribbon structure of the hexamer, the phosphate atoms (tan), and the O21 and O22 atoms of the glycerol (black). Also shown are the Cα atom of V12 (yellow), Q15 (blue), V18 (orange), E22 (purple), and K28 (red) [70].

**Figure 3 molecules-28-07080-f003:**
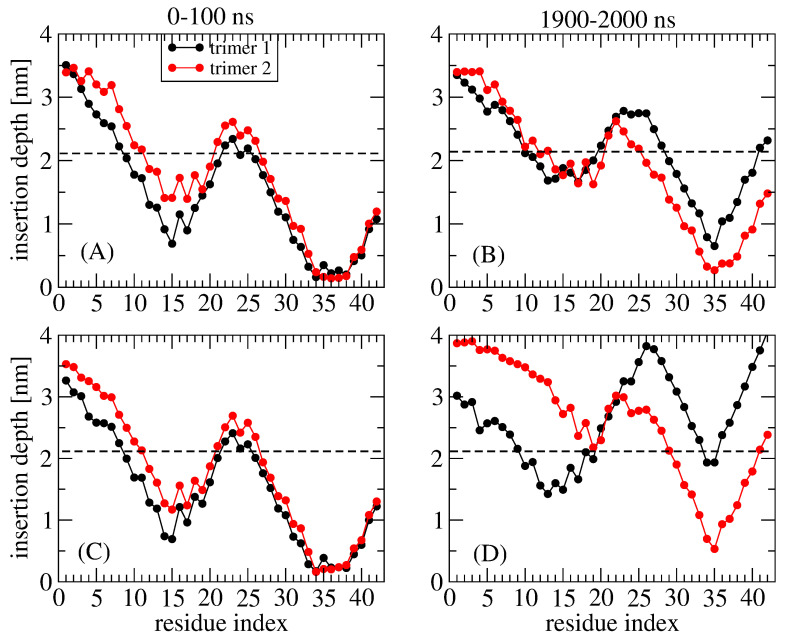
MD-generated insertion depths of the residues belonging to trimer 1 and trimer 2 in two time intervals: 0–100 ns and 1900–2000 ns. (**A**,**B**) MD run 1 at 303 K and (**C**,**D**) MD run 2 at 310 K [70]. The three chains in aqueous solution at the upper leaflet of the membrane are referred to as trimer 1 and the other three chains at the lower leaflet of the membrane are referred to as trimer 2.

**Figure 4 molecules-28-07080-f004:**
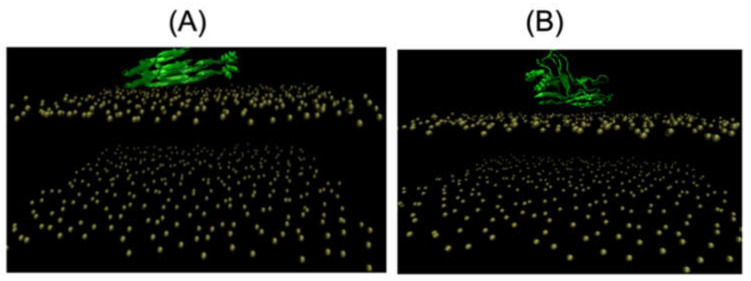
The two initial tau 306-378 dimer conformations. In both panels, the phosphate atoms of the lipid heads are in tan, and we show the van der Waals representation of the side chains of F378 residue in green. (**A**) Initial cryo-EM state. (**B**) Initial globular compact state [77].

**Figure 5 molecules-28-07080-f005:**
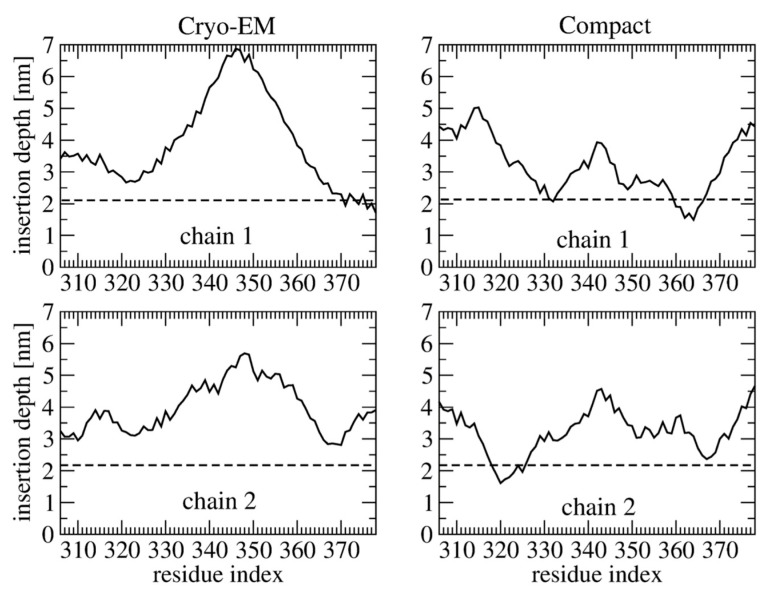
Average insertion depth of each residue of the two tau 306–378 chains using 1950−2000 ns of the MD simulation starting from the cryo-EM structure (**left**) and the globular compact structure (**right**) [77].

## Data Availability

Not applicable as this is a review article.

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
