# Peer review of "Recent Computational Advances Regarding Amyloid-β and Tau Membrane Interactions in Alzheimer’s Disease"

_molecules, 2023, doi:10.3390/molecules28207080_

Round 1

Reviewer 1 Report

The review entitled "Recent Computational Advances of Amyloid-β and Tau Membrane Interactions in Alzheimer’s Disease" covers a topic of great scientific interest, as amyloid peptides and the Tau protein play an important role in the pathogenesis of Alzheimer's disease. The review focuses on the computational studies of some key aspects, including the effect of different lipids on the interaction of Abeta 40-42 with the lipid bilayer to clarify the mechanisms with the cell membrane; the effect of different lipids on the secondary structure of peptides. The review is well structured, however, the reviewer suggests improvements to some sections.

To give a broader overview of the topic, the first part of section 3 "Detergent Effect of Amyloid-β. " (lines 218-222) should be enriched by reporting the results (albeit reported in brief) of the various studies on the detergent effect of amyloid peptides.   

In section 4 "Amyloid-β Pore Formation " the authors discuss the ability of amyloid peptides to form pores in the lipid bilayer. This section should be implemented with the results of studies on the incorporation of oligomers into the lipid bilayer and channel formation. In the literature, several studies show the incorporation of oligomers into the membrane and pore formation about both the lipid composition of the bilayer and the size of the oligomers. I suggest the authors report the results of the works indicated below as they show the interaction of Abeta 42 oligomers and the formation of the channel in cellular membranes and in planar lipid membranes. The formation of oligomers has been studied by means of different techniques. The reviewer thinks that the two works can improve the quality of the manuscript.

-Bode DC et al 2017;

-Meleleo D et al 2020.

The Introduction section presents a comprehensive overview of the distinct chemical, chemical-physical, and structural aspects of A40 and A42 examined using various analytical techniques.

Section 2, titled "Adsorption and Insertion of Amyloid-β," provides an in-depth account of the latest findings on the interaction between A40-42 and the membrane, as revealed by computational studies from 2020 onwards. Additionally, some studies conducted prior to 2020 have also been included. This section is deemed satisfactory.

Studies of Tau protein oligomer interactions with membranes are scarce. Section 5 reports on the most important studies about Tau-membrane interactions.

Author Response

We thank the referee for his/her comments. The referee does not bring any comments and suggestions for the introduction, section 1, section 2, section 5 and conclusion.

As stated in the manuscript, the main focus of this review is to discuss very recent computational advances. It is to be noted however that we have discussed various experimental aspects and the most recent findings regarding the interactions of amyloid-beta and tau proteins with membrane in the introduction. So for clarity, we prefer not to modify the content of sections 3 and 4 and we did not include the studies showing the incorporation of oligomers into the membrane and pore formation about both the lipid composition of the bilayer and the size of the oligomers (Bode DC et al 2017; Meleleo D et al 2020). Rather we added our Reference 2 at the beginning of the two sections.

Kind regards,

Professor Derreumaux

Reviewer 2 Report

The manuscript by P.H. Nguyen and P. Derreumaux, titled "Recent Computational Advances of Amyloid-β and Tau Membrane Interactions in Alzheimer's Disease," provides a comprehensive review of the latest advancements in computer modeling of the interactions between the Aβ peptide and tau protein with cell membranes. The authors also examine simulations of the interaction between free lipids and the Aβ peptide, which is an important aspect in understanding the mechanisms of amyloid formation.

Overall, this is an interesting and original review that can be valuable to both experts in computer modeling and biologists studying the interactions and effects of Aβ and tau protein on lipids and cell membranes.

Major:

1. The manuscript's writing style is my most significant concern, primarily due to the extensive sections of copied text from other articles, including the authors' own work. The similarity index of 39% indicates a high level of overlap. While the authors have appropriately referenced the sources, it is strongly advised to extensively revise the text to enhance its originality. Alternatively, when utilizing direct fragments of text, it is recommended to enclose them in quotation marks to clearly indicate their origin.

2. The conclusion of the manuscript could be improved. It should elaborate on the major challenges and gap areas identified in the field and discuss the corresponding future prospects. Additionally, it should aim to provide a clear and concise take-home message for the readers.

Minor:

3. In a paper, it is customary to first specify the primary author of the work, followed by "et al." to include all the other authors. Alternatively, if the work was supervised by a corresponding author, their name should be indicated. In this manuscript, there are instances of incorrect author mentions, where the supervisor of the work is listed instead of the primary author (e.g., Alonso et al.; Chang et al.; Strodel et al.; Nguyen et al.). These references need to be corrected in order to comply with accepted ethical standards.

4. Line 18: Replace "Aβ proteins" with the commonly used term "Aβ peptides".

5. In line 217, the section title "3. Detergent Effect of Amyloid-β" should be indicated in bold font, not italics.

6. In line 237, part of the compound name "1,2-dimyristoyl-sn-glycero-3-phosphocholine" is italicized, which should be corrected

7. Lines 253-255: The presented text has a different font than the rest of the text.

8. In line 257, the reference to Figure 1 should be in regular font, not bold.

9. Line 303: Remove parentheses around "Martini".

10. Ensure consistency in italicizing the term "in vitro" (see lines 36 and 165).

11. There are some missing sections that should be included in the article. These include "Author Contributions," "Funding," "Institutional Review Board Statement," "Informed Consent Statement," "Data Availability Statement," and "Conflicts of Interest." 

12. It is necessary to carefully review all references and align them with the journal's requirements. For example, in all references, the volume number should not be italicized. Additionally, the year should be in bold font, which is missing in references 39 and 59. Moreover, reference 7 is missing the year altogether.

13. Lastly, it would be beneficial to include a list of abbreviations in the article for clarity and ease of understanding.

Author Response

We thank the referee for his/her comments and finding our article as an interesting and original review.

We have addressed most major and minor aspects suggested by the referee and corrected the whole manuscript accordingly.

Best wishes

Prof. Derreumaux

Reviewer 3 Report

The manuscript “Recent Computational Advances of Amyloid-β and Tau 2 Membrane Interactions in Alzheimer’s Disease” presents a review on recent computational advances (starting from 2020) in the interaction of Aβ40 and Aβ42 alloforms and tau isoforms with cell membranes. The review is well written and gives a valuable contribution the researchers working in this field. I have no real criticism and recommend the paper to be accepted. I would perhaps include figures adapted from the literature of one or two examples of structural models (for example pore forming structures).

Author Response

We thank the referee with his/her very positive comments.

We did not include figures adapted from the literature of one or two examples of structural models (for example pore forming structures) because the amyloid-beta stuctures forming pores are very heterogeneous, consist of a wide range of conformations and oligomer sizes.

Round 2

Reviewer 1 Report

The review titled "Recent Computational Advances of Amyloid-β and Tau Membrane Interactions in Alzheimer's Disease" has been improved and the Reviewer provided no comments.

Reviewer 2 Report

In general, the authors have taken into account most of my comments. I would also like to note that the authors have made revisions to the text, reducing the similarity index. However, it still remains relatively high, exceeding 30%. I am ready to accept the article in its current form if the similarity index meets the requirements of the Molecules journal.